# An RNA-Scaffold Protein Subunit Vaccine for Nasal Immunization

**DOI:** 10.3390/vaccines11101550

**Published:** 2023-09-29

**Authors:** Joy-Yan Lam, Wan-Man Wong, Chun-Kit Yuen, Yau-Yee Ng, Chun-Hin San, Kwok-Yung Yuen, Kin-Hang Kok

**Affiliations:** 1Department of Microbiology, Li Ka Shing Faculty of Medicine, The University of Hong Kong, Hong Kong, China; joyyan@connect.hku.hk (J.-Y.L.); louisewong@hku.hk (W.-M.W.); jackyuen@connect.hku.hk (C.-K.Y.); yauyee@hku.hk (Y.-Y.N.); chsan@hku.hk (C.-H.S.); kyyuen@hku.hk (K.-Y.Y.); 2Centre for Virology, Vaccinology and Therapeutics, Hong Kong Science and Technology Park, Hong Kong, China; 3State Key Laboratory for Emerging Infectious Diseases, The University of Hong Kong, Hong Kong, China; 4AIDS Institute, Li Ka Shing Faculty of Medicine, The University of Hong Kong, Hong Kong, China

**Keywords:** protein subunit vaccine, SARS-CoV-2, self-adjuvanted, nasal vaccine, RNA scaffold, mucosal immunity

## Abstract

Developing recombinant proteins as nasal vaccines for inducing systemic and mucosal immunity against respiratory viruses is promising. However, additional adjuvants are required to overcome the low immunogenicity of protein antigens. Here, a self-adjuvanted protein-RNA ribonucleoprotein vaccine was developed and found to be an effective nasal vaccine in mice and the SARS-CoV-2 infection model. The vaccine consisted of spike RBD (as an antigen), nucleoprotein (as an adaptor), and ssRNA (as an adjuvant and RNA scaffold). This combination robustly induced mucosal IgA, neutralizing antibodies and activated multifunctional T-cells, while also providing sterilizing immunity against live virus challenge. In addition, high-resolution scRNA-seq analysis highlighted airway-resident immune cells profile during prime-boost immunization. The vaccine also possesses modularity (antigen/adaptor/RNA scaffold) and can be made to target other viruses. This protein-RNA ribonucleoprotein vaccine is a novel and promising approach for developing safe and potent nasal vaccines to combat respiratory virus infections.

## 1. Introduction

Nasal vaccinations have been an attractive immunization strategy in recent years due to the activation of mucosal immunity, the essential first line of defense against respiratory virus infections. There are various types of nasal vaccines for different viral infections such as influenza, COVID-19, and respiratory syncytial viruses (RSV). Mucosal secretory IgA and neutralizing antibodies have been recognized as pivotal for preventing the entry of pathogens [1,2]. After the COVID-19 pandemic, safe and effective nasal vaccines are in even higher demand among clinicians and researchers as they can possibly prevent potential emerging pandemics. One of the highly promising types of vaccines is protein subunit vaccines. Due to their simplicity, ease of manufacturing, and high safety profiles, several pre-clinical candidate vaccines are in clinical trials for further testing [3,4].

However, the greatest pitfall of protein subunit vaccines is the extremely low immunogenicity in vivo [4]. The use of adjuvants is almost always essential when considering protein vaccines. Nevertheless, nasal protein vaccines are scarce, possibly due to obstacles in selecting proper adjuvants that can balance the effectiveness and side effects in the nasal cavity, which is prone to allergic reactions [5]. Chemical adjuvants that succeeded in intramuscular vaccination, such as Alum/CpG, must be carefully evaluated for their use in intranasal vaccination.

Immunostimulative RNAs are a class of safe adjuvants that act as TLR agonists to trigger innate immune pathways, leading to chemokine production and producing an effective adaptive immune response. RNA sensing TLR signaling pathways involve protective interferons and chemokine production, leading to a Th1 or balanced immune response [6,7]. Like recombinant proteins, single-stranded (ss) or double-stranded (ds) RNAs are versatile and can be manufactured without complicated procedures. Poly(I:C), the dsRNA adjuvant, has undergone several clinical trials recently, acting as an intramuscular immunostimulant [8]. Clinical studies suggest that Poly(I:C) is tolerable in humans [9]. Regarding protein vaccines or nasal administrations, RNA as an adjuvant remains largely unexplored. Especially in the nasal mucus lining, RNAs are more prone to degradation. As a result, exploring innovative strategies to combine proteins and RNA for nasal delivery can provide new insights for protein nasal vaccines to respond to the ongoing global demand.

Using SARS-CoV-2 as a model, we have here created a self-adjuvanted ribonucleoprotein nasal vaccine that effectively induced strong systemic and mucosal immunity in the pre-clinical mouse model. The ribonucleoprotein vaccine, WNPRBD-R266, is a protein-RNA complex that formed nano-sized particles and both the protein (WNPRBD; adaptor-antigen) and RNA (R266; adjuvant and RNA scaffold) components are originated from SARS-CoV-2, with defined sequences. As an immunostimulative ssRNA, R266 induces strong interferon and chemokine production, while keeping inflammatory responses minimal, implying minimal irritations. Without any external chemical adjuvants, WNPRBD-R266 was able to elicit antibody responses in both sera and BALF and robust antigen-specific T cell responses. Vaccinated mice also obtained sterilizing immunity against live SARS-CoV-2 virus challenge. Furthermore, we utilized single-cell RNA sequencing (scRNA-seq) and obtained high-resolution data on BALF-flushed cells to evaluate lung-resident immune cells in high detail. Macrophages, B lymphocytes, and T lymphocytes were highly activated. Importantly, the scRNA-seq data highlighted the importance of a prime-boost regimen for primary vaccination as the boost dose has brought about the most drastic change in the lung-resident immune cell population. In summary, the WNPRBD-R266 ribonucleoprotein vaccine is a candidate nasal vaccine that does not require external adjuvants. With its “all-virus” design, WNPRBD-R266 has great potential and warrants further clinical evaluations.

## 2. Materials and Methods

### 2.1. Plasmids, Cell Cultures and Viruses

DNA plasmids expressing codon-optimized SARS-CoV-2 spikes and nucleoprotein were gene-synthesized by (Sangon Biotech, Shanghai, China). Expression plasmids for protein expression were designed, cloned, and sequenced using standard cloning methods. VeroE6 (CRL-1586) was obtained from ATCC and cultured according to the supplier’s instructions. VeroE6-TMPRSS2 cells were obtained from the Japanese Collection of Research Bioresources (JCRB) cell bank and cultured in DMEM medium. Ancestral SARS-CoV-2 virus HKU-001a (GenBank: MT230904) and B.1.351/Beta variant virus (GenBank: OM212470) were cultured, titered, and plaque-purified in VeroE6-TMPRSS2 cells [10,11]. All in vitro and in vivo experiments involving SARS-CoV-2 viruses were performed in a BSL-3 laboratory according to approved standard operating procedures at the Department of Microbiology, HKU.

### 2.2. Protein Production, Expression and Purification

The expression plasmid for WNPRBD fusion was constructed by cloning. Ancestral SARS-CoV-2 (MT230904) full-length nucleoprotein was fused, without protein linker, to spike RBD (306–543aa) by PCR and cloned into a CMV-driven expression vector, with N-terminal human tissue plasminogen activator signal peptide (MDAMK RGLCC VLLLC GAVFV SPSAA), 6×His tag, and a HRV 3C protease cleave site. Plasmids were transfected into Expi293F cells (Gibco^®^, ThermoFisher Scientific, Inc., Waltham, MA, USA) for protein production using Expifectamine 293 transfection reagent following the manufacturer’s instructions. Culture supernatants were harvested at 96 h post-transfection, filtered, and passed through Ni Sepharose Excel resins (Cytiva, Marlborough, MA, USA) for purification. Eluted proteins were further purified by size exclusion chromatography Superdex S200 column (Cytiva, Marlborough, MA, USA). Protein products were buffer exchanged to standard 1× PBS (pH 7.4) with 10% (*v*/*v*) glycerol and concentrated using the Pierce Protein Concentrator (ThermoFisher Scientific, Inc., Waltham, MA, USA). For WNPRBD, the A260/A280 ratio was kept below 0.6 to ensure minimal nucleic acid contamination. For other recombinant proteins, ancestral SARS-CoV-2 spike RBD proteins were also expressed in Expi293F using plasmid transfection and affinity purified by His-tag. Nucleoproteins were expressed in the bacteria system and affinity purified by His-tag.

### 2.3. T7 In Vitro Transcription

R20 and R266 RNA were produced using the HiScribe T7 Quick High Yield RNA Synthesis Kit (New England Biolabs, Ipswich, MA, USA) following the manufacturer’s instructions. The template for R20 was produced by oligo annealing. Sense and antisense oligos with T7 promoter were obtained from Integrated DNA Technologies, Inc. Coralville, Iowa, USA, annealed in a thermocycler, and used directly in T7 transcription. For R266, the template sequence was amplified from viral cDNA by PCR and subcloned into pMD19-simple (Takara Bio, Inc., Kusatsu, Shiga Prefecture, Japan). Purified and linearized plasmids were then used in T7 transcription. The resulting ssRNA products were treated with DNase I, purified using the RNAiso Plus (Takara Bio, Inc., Kusatsu, Shiga Prefecture, Japan) reagent and precipitated with isopropanol. All RNAs were dissolved in a standard 10 mM Tris–1 mM EDTA buffer.

### 2.4. In Silico Analysis and Structure Prediction

Protein structure prediction was performed using the Robetta web server with the RoseTTAFold model [12]. RNA secondary structure prediction was performed using the RNAfold web server and visualized by ViennaRNA. RNA 3D structure prediction was performed using RNAComposer [13,14].

### 2.5. Nanoparticle Tracking and Measurement

Nanoparticle tracking analysis was performed on ZetaView Twin (Particle Metrix, Ammerse, Germany). Briefly, the system was calibrated with PS100 polystyrene standard beads. Camera sensitivity and contrasts were calibrated. Samples were diluted in filtered MilliQ water and injected into the system. Particle concentrations, particle sizes and respective zeta potentials were measured and reported by ZetaView software version 8.05.11.

### 2.6. Animal Immunizations and Infections

All animal experiments were approved by the Committee on the Use of Live Animals in Teaching and Research of the University of Hong Kong (CULATR 5108-19; approval date: 24 March 2021). Female BALB/c mice of 8–10 weeks were immunized intranasally under anesthesia. Each dose of ribonucleoprotein vaccine comprised 12 µg of WNPRBD and respective amount of R266 (or R20) as described per experiment. The vaccine was prepared by mixing of purified WNPRBD protein and R266 (or R20) and incubating at room temperature for 15 min. The vaccine was then placed on ice until vaccination. Final vaccine volume was controlled at 20 µL and buffer composition comprised 1× PBS, 3% glycerol, 5 mM Tris, 0.5 mM EDTA. Blood was collected from the facial vein at indicated timepoints. Animals were sacrificed by an overdose of anesthesia at the experiment endpoint. Bronchoalveolar lavage fluid (BALF) was collected by inserting a catheter in the trachea of the euthanized mice, followed by instilling PBS into the bronchioles and lung. The collected fluid was centrifuged and the resulting cell pellets and supernatant were collected for downstream analysis. For animal infections, 8–10 weeks old BALB/c mice were first immunized with 2 doses of 12 µg WNPRBD-25 µg R266, with 14 days in between. At 7 days before infection, mice were boosted with 1 dose of 12 µg WNPRBD-25 µg R266. Vaccinated mice were then intranasally inoculated with 20 µL B.1.351/Beta viruses at 1 × 10^5^ PFU/mouse. Mice lungs and nasal turbinates were harvested and homogenized by the Tissue Lyzer II (QIAGEN, Inc., Hilden, Germany) system in PBS for downstream analysis.

### 2.7. RNA Gel Shift Assay

Purified WNPRBD protein at varying amounts (0–2 µg) and R266 ssRNA at a fixed amount (100 ng) were mixed at room temperature in buffer composition identical to that used in animal immunization. After 15 min incubation, samples were split into two equal portions. One portion was loaded into 8% SDS-PAGE for Western blot protein analysis using anti-His antibody (Abcam plc, Cambridge, UK) and the blot was visualized using a WesternBright ECL (Advansta, Inc., San Jose, CA, USA) and a Sapphire Biomolecule Imager (Azure Biosystems, Inc., Dublin, CA, USA). The other portion was loaded into 4% native PAGE/0.5× TBE and stained by GelRed (Biotium, Inc., Fremont, CA, USA) for RNA detection. The resulting native gel was visualized by an Azure 200 Gel Imager.

### 2.8. Fluorescence Polarization Assay

Varying concentrations (0.0012–2.5 µM) of purified WNPRBD, full-length nucleoprotein, and spike RBD were incubated with 100 nM 5′-HEX labeled R20 oligo (synthesized by Integrated DNA Technologies, Inc., Coralville, IA, USA) for 30 min at room temperature, in buffer composition identical to that used in animal immunization. All incubation was carried out in black, opaque 96-well microplate (SPL Life Sciences, Gyeonggi-do, Republic of Korea). Fluorescence polarization data was captured using a Biotek Cytation 5 Multimode Reader (Agilent Technologies, Inc., Santa Clara, CA, USA). The dissociation constant in the specific buffer system was analyzed and calculated using a GraphPad Prism 9 built-in specific binding model.

### 2.9. Enzyme-Linked Immunosorbent Assay (ELISA)

High-binding 96-well microplates (Corning 3690) were coated with 2 µg/mL recombinant proteins using a 0.05 M sodium bicarbonate buffer (pH 9.6) at 4 °C overnight. The next day, microplates were washed and blocked with a casein buffer. Samples were serially diluted in PBS and added to microplates for incubation. Antigen-specific antibodies were detected by HRP-conjugated secondary antibodies to mouse IgG (Abcam plc, Cambridge, UK) or mouse IgA (Invitrogen, Corp., Carlsbad, CA, USA). For IgG subclasses, biotinylated secondary bodies and streptavidin-HRP were used (BioLegend, Inc., San Diego, CA, USA). 1-Step Ultra TMB-ELISA Substrate Solution (ThermoFisher Scientific, Inc., Waltham, MA, USA) was used for signal generation. Absorbance at 450 nm was then measured using a Vatioskan LUX multimode microplate reader (ThermoFisher Scientific, Inc., Waltham, MA, USA) with SkanIt Software version 6.1.0.51. Titers were calculated in GraphPad Prism 9 by performing 4-parmeter logistical fitting with absorbance data. Antibody endpoint titers were determined as the interpolated reciprocal of the dilution having the same absorbance as the mean of blank wells plus 10 standard deviations.

### 2.10. Fluorescent Reduction Neutralization Assay (FRNT)

Neutralization assays were performed with live ancestral SARS-CoV-2 viruses in a biosafety level 3 laboratory. VeroE6 cells were seeded in 96-well black plates (SPL Life Sciences, Gyeonggi-do, Republic of Korea). Samples were serially diluted in plain medium and 1000 PFU/well viruses were added for incubation at 37 °C for 1 h. Sample–virus mixtures were then added to the seeded VeroE6 cells and incubated for 1 h at 37 °C. Cells were washed with PBS and replenished with culture medium containing 1% FBS. Inoculated cells were further incubated for 6 h, and then were fixed with 4% formalin. The plates were washed, permeabilized with 0.1% NP40, blocked with 2% BSA, and stained with in-house rabbit anti-SARS-CoV-2 nucleoprotein polyclonal antibody and detected by anti-rabbit Alexaflour 488 (Abcam plc, Cambridge, UK). Fluorescent positive cells were detected by a Biotek Cytation 7 Cell Imaging Multi-Mode Reader (Agilent Technologies, Inc., Santa Clara, CA, USA) with data captured by Gen5 Image Prime version 3.11.19. Neutralization titers were calculated and determined in GraphPad Prism 9 by performing 4-parameter logistical fitting on the detected foci. The 50% focus reduction neutralization titer (FRNT50) was determined as the interpolated reciprocal of the dilution having 50% reduction in infected fluorescent loci compared to control wells.

### 2.11. Reverse Transcription and Quantitative PCR

Viral RNAs were extracted using QIAamp viral RNA (Qiagen) and reverse transcriptions were performed using a PrimeScript RT reagent kit with a gDNA Eraser (Takara Bio, Inc., Kusatsu, Shiga Prefecture, Japan). Viral RNA in animal samples were quantified by QuantiNova Probe RT-PCR kit (QIAGEN, Inc., Hilden, Germany) using primers targeting RNA-dependent RNA polymerase (RdRp). Taqman probe and qPCR primers were adopted from previous studies [10].

### 2.12. Bead-Based Cytokine Assay

Protein cytokines (mIFNβ, mIL6, mCXCL10, mCCL5, mIL1b, and mTNFα) were quantitated using the LEGENDplex multi-analyte flow assay kit (Mouse Anti-Virus Response Panel, BioLegend, Inc., San Diego, CA, USA, 740621). Briefly, bronchoalveolar lavage fluid (BALF) was collected by inserting a catheter in the trachea of the euthanized mice, followed by instilling PBS into the bronchioles and lung. The collected fluid was centrifuged, and resulting cell pellets and supernatant were collected for multi-analyte flow assay following the manufacturer’s instruction. A standard curve of each protein cytokine was plotted for the calculation of the protein concentration of each cytokine in BALF using LEGENDplex Data Analysis Software (BioLegend, San Diego, CA, USA).

### 2.13. T Cell Activation-Induced Marker (AIM) Assay

Groups of mice were mock-vaccinated (PBS) and vaccinated with WNPRBD and WNPRBD-R266 as indicated. BALF and lung were harvested at 7 days post-second vaccination. Cells derived from BALF and lung-disassociated cells were used for the T cell activation assay. Cells were counted and activated by the peptide pool (PepMix SARS-CoV-2 S-RBD; JPT Peptide Technologies, Berlin, Germany) overnight. Cells were then stained with antibodies including anti-mouse CD8a-APC-Fire750 (Rat IgG2ak; 0.2 mg/mL; Biolegend 100766) 1:400; anti-mouse CD3-SB780 (Rat IgG2bk; 0.2 mg/mL; eBioscience 78-0032-82) 1:100; anti-mouse IL2-PE (Rat IgG2bk; 0.2 mg/mL; Biolegend 503808) 1:100; anti-mouse TNFα-BV421 (Rat IgG1k; 0.2 mg/mL; Biolegend 506328) 1:200; anti-mouse IFNγ-APC (Rat IgG1k; 0.2 mg/mL; Biolegend 505809) 1:100; anti-mouse CD4-BB700 (Rat IgG2ak; 0.2 mg/mL; BD 566408) 1:400. Data were collected by LSRFortessa (BD Biosciences, Franklin Lakes, NJ, USA) and analyzed using FlowJo version 10 (BD Biosciences).

### 2.14. Histology and Immunohistology Staining

Animal lung tissues were collected and fixed with 10% formalin for 24 h and paraffin-embedded. Three sections from each animal were used for histology analysis. Upon staining, sections were mounted onto slides and dewaxed in xylene. Antigen retrieval was performed by autoclaving slides at 121 °C for 3 min in Vector antigen unmasking solution, citrate-based (Vector Laboratories, Inc., Newark, CA, USA). Sections were then permeabilized with 0.1% Triton X-100, quenched by Sudan Black B, and blocked with 1% BSA. Infected cells were stained by in-house rabbit anti-SARS-CoV-2 nucleoprotein polyclonal antibody and detected by anti-rabbit Alexaflour 488 (Abcam). Cell nuclei were stained with Hoechst 33258 (Thermo). For H&E staining, tissue sections were stained with Gill’s haematoxylin and eosin-Y. Images were acquired using the Olympus BX53 light microscope (EVIDENT, Tokyo, Japan).

### 2.15. Single-Cell RNA Sequencing Sample Preparation and Processing

BALF-flushed cells were used for scRNA sequencing. BALB/c mice were immunized according to the experimental setting as described in the main text. Mice BALFs were harvested, centrifuged, and the resulting cell pellets were resuspended in PBS/1%FBS. The 10X Chromium library preparation and sequencing were performed at the Centre for PanorOmic Sciences, HKU. Briefly, single-cell encapsulation and cDNA libraries were prepared by Chromium Next GEM Single Cell 5′ Reagent kit v2 (Dual Index) and Chromium Next GEM Chip K Single Cell kit according to the manufacturer’s instructions. Cell suspensions were counted and viabilities were assessed. Input cell number was then normalized to 22,750 cells across samples. Cells were loaded into 10X Chromium Single Cell chip. Reverse transcriptions, cDNA cleanup and amplification were performed on Gel Beads-in emulsions, followed by fragmentation, adapter ligation and index PCR. Library size distributions were determined by Agilent 2100 Bioanalyzer. Sequencing was performed using Illumina NovaSeq 6000 for Pair-End 151 bp sequencing, and sequencing reads data were imported to 10X CellRanger pipeline for data preprocessing and initial filtered feature barcodes and matrices were exported.

### 2.16. Single-Cell RNA Sequencing Data Processing and Automated Cell Type Identification

Based on the filtered barcodes and matrices generated from 10X CellRanger, the sequencing data were further analyzed using Seurat v3 on R 4.3.0. Briefly, all three single-cell datasets with their individual original identity were merged in Seurat and reanalyzed. Data was filtered, normalized, scaled, dimensionality reduced and clustering was performed. Merged data can be distinguished based on the assigned original identity. For automated cell type identification, the open-source package SingleR [15] was used together with the ImmGen reference dataset through the celldex package. Cell types with less than 100 labeled cells are considered ambiguous and were discarded in downstream analysis. Differential gene expression between clusters and cell types was determined by Wilcoxon rank sum test in Seurat. Cell type identity was validated by evaluating gene markers. For further analysis, barcodes of specific cell types from Boost sample were extracted using Seurat subset function and reclustering was performed to have higher resolution clusters for cell annotation within a major cell type. UMAP and dotplots were generated and visualized using Seurat and ggplot2 functions.

### 2.17. Statistical Analysis

Statistical significances were calculated and analyzed using GraphPad Prism v9.5.0.

## 3. Results

### 3.1. WNPRBD Fusion Protein Forms Nano-Sized Complex with R266 ssRNA

Protein subunit vaccines are highly promising due to their high safety profiles but they generally lack immunogenicity. We, therefore, designed a new vaccine technology to co-deliver the major neutralizing antigen and immunostimulative RNA for effective vaccination. To associate the two parts together, we utilized the natural RNA binding ability of viral structural proteins as the adaptor and designed the WNPRBD (Whole NucleoProtein-spike RBD) fusion protein comprising SARS-CoV-2 spike RBD directly fused with SARS-CoV-2 nucleoprotein (Figure 1A). Recombinant proteins were expressed and purified in the mammalian cell system to ensure proper post-translational modifications (Figure 1B). A260/A280 ratio was kept below 0.6 to ensure minimal RNA contamination. For the immunostimulative RNA and RNA-scaffold, we aimed to produce short, non-coding ssRNA that forms high order secondary structure by general in vitro transcription. We, therefore, chose the 5′UTR of the SARS-CoV-2 genome, at 265 nt and with a defined sequence and suggested secondary structures. With modifications designed for T7 in vitro transcription, a 266 nt ssRNA with 5′ triphosphate was generated and termed R266. Secondary and tertiary structures prediction were performed (Figure 1C,D) and the T7 transcription product size was validated by Urea-PAGE (Figure 1E). Due to the natural properties of nucleoprotein, WNPRBD proteins and R266 ssRNA were mixed at room temperature with defined buffer to form protein-RNA complexes. The conceptual WNPRBD-R266 ribonucleoprotein complex is illustrated in Figure 1F.

To characterize the WNPRBD-R266 complex, the particle size was examined (Figure 1G). Nanoparticle tracker analysis determined that particles were formed with diameter 135.85 nm, and a surface charge (Zeta potential) of −25.61 mV. This suggested that WNPRBD binds to R266 ssRNA and forms nano-sized complex, and the negative surface charge was presumably due to the RNA phosphate backbone. R266 by itself did not form any detectable particles, while larger particles at 187.70 nm diameter were formed by WNPRBD protein alone, with a +12.01 mV surface charge. This indicated that in the absence of R266, WNPRBD retained oligomerization properties of nucleoprotein, and its overall charge referred to protein surface charges. We further validated the RNA binding properties by gel shift assay and fluorescence polarization assay. In non-denaturing PAGE, WNPRBD formed complexes with R266 (lanes 7–8, Figure 1I). At lower concentrations of WNPRBD, R266 was in excess, forming smaller aggregates (lanes 2–6, Figure 1I). Using the fluorescence polarization assay, WNPRBD exhibited comparable dissociation constants as full-length nucleoprotein in the assay buffer system (Figure 1J), indicating the strong binding between WNPRBD and the RNA. These data clearly demonstrated that the WNPRBD protein (adaptor-antigen) maintains the RNA binding ability of natural SARS-CoV-2 nucleoproteins and forms ribonucleoprotein complexes with R266 RNA. The WNPRBD-R266 complex therefore contains the major neutralizing antigen, RNA-binding nucleoprotein, and immunostimulative scaffold RNA, which acts as a vaccine as a whole.

### 3.2. WNPRBD-R266 Ribonucleoprotein Is Immunogenic and Induces Mucosal IgA with Minimal Release of Inflammatory Cytokines

To evaluate the vaccine potential of the WNPRBD-R266 ribonucleoprotein complex, mice immunization experiments were set up (Figure 2A). Mice were given three doses of WNPRBD-R266 intranasally. Sera and bronchioalveolar lavage fluids (BALFs) were harvested to determine systemic and mucosal antibody responses, respectively. R20 is a 20 nt ssRNA with 5′ triphosphates derived from the SARS-CoV-2 nucleoprotein sequence (63–81 nt) and was included as a comparison control. Surprisingly, only WNPRBD-R266 complexes, but not WNPRBD-R20, elicited mucosal antigen-specific IgA in BALF (Figure 2B). WNPRBD-R266 elicited higher mucosal IgG (Figure 2C) as well as serum systemic IgG (Figure 2D) than WNPRBD-R20. This indicated that the length of RNA may influence the immunogenicity of the ribonucleoprotein vaccine. Hence, we tested whether different lengths of ssRNA (both with 5′ triphosphates) may affect their immunostimulative potential in the WNPRBD-RNA complex. WNPRBD-R266 or -R20 complexes were intranasally administered to mice and BALF samples (fluid and cells) were harvested at 16 h post-administration to determine cytokine production (as illustrated in Figure 2E). First, we quantitated cytokine transcript expressions in BALF-flushed cells (Figure 2F). Both R266- and R20-containing protein complex induced strong interferon responses (mIFNβ, mIFITM3 and mRIG-I) and chemokine signals (mCXCL10 and mCCL5), but minimal inflammatory cytokines such as mIL6, mIL1β and mTNFα (Figure 2F). Second, we validated the cytokine production at the protein level in BALF using a bead-based assay (Figure 2G). We observed potent induction of chemokine mCXCL10, and consistently minimal induction of inflammatory cytokines. It is noted that the unsuccessful detection of IFNβ may be due to the high protein turnover rate in BALFs. These data indicated that WNPRBD-R266 or -R20 complex are both sufficiently immunostimulative and act as a self-adjuvanted nasal vaccine in mice, with minimal induction of inflammatory cytokines. However, only WNPRBD-R266, but not -R20, induced a significant mucosal IgA response (Figure 2B). This prompted us to focus solely on further investigating the WNPRBD-R266 vaccine efficacy.

### 3.3. Complex Formation of WNPRBD-R266 Is Essential for Mucosal Immune Activation and Elicits a Balanced Immune Response

Next, we designed an experiment to examine the antibody response of different doses of R266 in mice (Figure 3A). Mice were intranasally immunized with two doses of WNPRBD-R266 at 5 µg/12.5 µg/25 µg. Sera and BALFs were harvested for the examination of antibody responses. Systemically, all three doses of R266 showed similar antigen-specific IgG responses (Figure 3B,E). At the mucosal level, three doses showed a dose-dependent antibody induction, where WNPRBD with 25 µg R266 showed the highest induction of mucosal IgG and IgA (Figure 3C,D and 3F,G, respectively). At 5 µg, the mucosal immune activation was marginal. Therefore, we chose WNPRBD-25 µg R266 as the optimal dose for vaccination and hence determined the antibody subclass. WNPRBD-R266 (25 µg) elicited a balanced Th1 and Th2 response as indicated by the ratio of IgG1 and IgG2b in mice (Sera, Figure 3H,I; BALF, Figure 3J,K). Furthermore, the FRNT neutralization assay showed that the antibodies in both sera and BALFs were functional and provided significant neutralizing capabilities against the live SARS-CoV-2 virus (Figure 3L,M).

To investigate whether the complex formation is essential for the immune responses elicited by WNPRBD-R266, we designed an experiment to compare WNPRBD-R266 and RBD-only-R266 (as illustrated in Figure 3N). We chose to use RBD-only-R266 as a negative control, rather than an RNA-binding defective mutant of WNPRBD, because of the technical difficulties in creating a SARS-CoV-2 nucleoprotein lacking RNA binding property. As expected, RBD-only R266 (mixture) was able to induce serum IgG and BALF IgG (Figure 3O,P), but not the mucosal BALF IgA (Figure 3Q). WNPRBD-R266 (complex) induced significant IgA in BALF (Figure 3Q).

Interestingly, the neutralizing ability of the antibodies elicited by RBD-only-R266 was very poor (Figure 3R,S). The immune response of RBD-only-R266 was also significantly skewed towards IgG1, while WNPRBD-R266 induced a response highly similar to the mRNA vaccine (1 µg BioNTech, intramuscular) in mice (Figure 3T). These data strongly suggested that WNPRBD-R266 elicited a functional and balanced immune response in mice and ribonucleoprotein complex formation is indispensable.

### 3.4. Robust Lung-Resident T-Cell Activation in WNPRBD-R266 Immunized Mice

To determine standard cell-mediated immune responses, mice were immunized with two doses of WNPRBD-R266 and BALF cell pellets were harvested at day 7 post-2nd dose for spike-specific T cell activation assay (as illustrated in Figure 4A). Spike-specific CD4+ (Figure 4B–D) and CD8+ (Figure 4E–G) T cell activation were observed. Bifunctional (mainly IFNγ and TNFα) and multifunctional (IFNγ, TNFα, and IL2) T-cells contributed more than one-third of the total spike-specific CD4+ or CD8+ cell activation (Figure 4H and 4I, respectively), suggesting robust lung T cell functions were induced. In summary, as evidenced by antibody induction and T cell activation, the WNPRBD-R266 ribonucleoprotein vaccine induced both humoral and cell-mediated adaptive immune responses.

### 3.5. WNPRBD-R266 Ribonucleoprotein Immunization Protects Mice from the SARS-CoV-2 Challenge

To evaluate the protective capabilities of WNPRBD-R266 against infection, mice were intranasally immunized with three doses of the WNPRBD-R266 vaccine and intranasally challenged with live SARS-CoV-2 B.1.351/Beta viruses (Figure 5A). Mice lungs and nasal turbinates were harvested at day 2 post-inoculation and analyzed by plaque assay, RT-qPCR, and histological staining. No viral titer was detected in both lung and nasal turbinate of WNPRBD-R266 vaccinated mice (Figure 5B and Figure 5C, respectively). Viral RNA copies were not detected in lung (Figure 5D) but were detected in 3 out of 4 mice in the nasal turbinate of vaccinated mice at day 2, albeit in lesser amounts than in non-vaccinated mice (Figure 5E). Consistently, no virus-positive cells were detected in immunohistological staining (Figure 5F). All three assays clearly showed that WNPRBD-R266 provided complete protection to the animal and suggested that sterilizing immunity is induced.

### 3.6. High-Resolution Profiling of Lung-Resident Immune Cells in WNPRBD-R266 Vaccinated Mice BALF by Single-Cell RNA Sequencing

To provide mechanistic insights on how WNPRBD-R266 elicits such a potent mucosal immune response, a single-cell RNA sequencing (scRNA-seq) experiment was performed. Mice were primed/boosted with WNPRBD-R266 and BALF-flushed cells were analyzed by scRNA-seq. Samples were harvested at 7 days pre-2nd dose (Prime) and 7 days post-2nd dose (Boost) (as illustrated in Figure 6A). Mock vaccinated mice (Naïve) were used for control. According to the number of differentially upregulated genes, the Boost had substantial changes while the Prime was not significantly different from the Naïve (Figure 6B). An UMAP reduction map of cell clusters also revealed the enrichment of several unique cell types in the Boost only, which include B cells, NK cells, NKTs, T cells and ILCs (Figure 6C–E).

To elucidate the subtle differences among cell clusters, barcodes of cells from specific clusters from the Boost were extracted and reclustered for higher resolution cell type identification based on differentially expressed genes. Macrophages, monocytes, and neutrophils (MMNs) reclustering showed a significant difference between the Prime and the Boost (Figure 6F) and cells were identified mainly by Fc receptor (FcR) expressions [16] (Figure 6G). Alveolar macrophages expressing Fcgr3 and Fcgrt were found. Cells with Fcgr2 and Fcgr3 were monocyte-DC signatures and were termed antigen-presenting monocytes. Both cell types had increased gene expression in the Boost sample and were activated. Proliferating macrophages expressing Hmgb2, Mki67, Top2a, Birc5 and Pclaf were found in both the Prime and Boost [17]. Cells with intermediate FcR expression were termed transitioning macrophages and were only found in the Prime. Monocyte-derived dendritic cells (MoDCs) expressing Fcgr1, Fcgr3 and Fcgr4 were only found in the Boost [16]. Moreover, groups of innate-like lymphoid cells were identified only in the Boost, expressing Cd3e, Cd3g and T lymphocytes markers such as Ccl6, Tl7r and Nkg7. There was also a small cluster expressing Cd8b1. The analysis of MMNs clusters suggested that the Boost dose was able to activate innate cells strongly, as reflected by the drastic differences between FcR expressions. Consistent with the previous cytokine analysis, there were minimal traces of inflammatory macrophages. Also, monocytes and neutrophils that normally induce inflammation during vaccination and infection only contribute to a small portion of the entire immune cell population profile in the mice BALF (Figure 6H). This again suggested that WNPRBD-R266 induces minimal inflammation.

We then reclustered and analyzed the B cells. The B cells were almost non-existent in the Naïve and Prime samples but were significantly enriched in the Boost. The Boost B cell recluster returned two major clusters, which were naïve B cells, with high expression of Fcmr, Ighm, and Ighd, and mature B cells that express Igha, Ighg1, and Ighg2b. (Figure 6I,J). Memory B cell markers such as Itgb1, Itgb4, Lgals1, Ly6c2, and Ms4a4b were also identified [18]. The mature B cells also carry plasma cell signatures such as Irf4 and Cd27, indicating that B cells transitioned to antibody-producing cells in Boost [19].

The T cell population was also reclustered and revealed several distinctive clusters (Figure 6K). Cell identities and functionality of these clusters were mainly evaluated by expression of Cd44, Il7r/Cd127 and Sell/Cd62L. Detailed differentially expressed genes between T cell clusters were examined (Figure 6L). Effector CD8 T cells (Cd44+/Il7r−/Sell-) expressing high levels of Cd8a, Cd8b1, Ccl5, and Nkg7 were identified. There were also clusters of naïve Cd8 T cells (Cd44−/Il7r+/Sell+). Recently activated T cells (Cd44−/Il7r+/Sell−) with elevated Klf2 expression and transitioning T cells (Cd44+/Il7r+/Sell−) were identified. These clusters expressed similar gene signatures and presumably underwent transition status under the boost dose effect. Inflammatory Cd8 T cells were identified with Gzmk expression [20]. Interestingly, innate-like T cells were also identified, with macrophage markers such as Lyz2 and Fcer1g.

## 4. Discussion

The study of ribonucleoprotein (RNP) complexes for intraperitoneal immunization was documented more than 30 years ago. Rabies virus RNP and G protein enclosed in liposome was proposed and can elicit protective responses in mice and raccoons, but RNP alone provided no protection [21]. Moreover, unadjuvanted rabies RNP was used for immunization in monkeys and successfully protected the animal, however, no neutralizing antibodies were found [22]. The major drawback of using viral RNPs directly is that the production process involves viral cultures and no defined compositions. In more recent studies, oligomerization of respiratory syncytial virus (RSV) nucleoprotein and bacterial RNA was used as a nasal vaccine in mice in the presence of bacterial endotoxins as adjuvant and elicited anti-N antibodies [23]. This indicates that protein-RNA complex, or ribonucleoprotein, has great potential as a vaccine, yet this potential is largely underexplored. 

At the time of this study, there are various vaccine platforms being developed for infectious disease, such as lipid nanoparticles (LNP), viral vectors, live attenuated virus and more. Each platform excels in different areas, such as ease of use, ability to activate cross-protection, as well as robustness of activation. LNP-mRNA vaccines are particularly well recognized due to the widespread usage during the COVID-19 pandemic, and have been implemented for influenza and respiratory syncytial virus (RSV) as well [24,25,26]. Among the vaccine candidates, several can enter clinical trials for further evaluation and become clinically available. For example, in more recent times, the FDA has approved the first RSV vaccine for people 60 years old and older, which is an adjuvanted protein vaccine [27]. However, there are still no protein vaccines, with or without adjuvants, available for human nasal vaccination.

With the growing demand for safer vaccines and the need for nasal vaccines to prevent future pandemics, we have rationally designed and produced the self-adjuvanted ribonucleoprotein nasal vaccine, WNPRBD-R266. By combining purified, mammalian expressed recombinant proteins with in vitro transcribed ssRNA, an RNP with known composition and sequences is produced. This greatly eliminates the unknowns of viral RNPs and a defined composition would facilitate clinical uses. Moreover, we designed WNPRBD-R266 RNP using only viral components, namely the SARS-CoV-2 spike RBD (antigen), SARS-CoV-2 nucleoprotein (adaptor), and SARS-CoV-2 5′UTR (adjuvant and RNA scaffold). This ensures the entire RNP is “all-virus” and eliminates unwanted responses. Furthermore, the WNPRBD protein contains both a major virus-neutralizing target and a potential cross-protection epitope, making it a dual-antigen design.

We first performed a basic characterization of WNPRBD-R266 (Figure 1). SARS-CoV-2 nucleoprotein is well-recognized for phase-separation in the presence of RNA [28,29,30] and its oligomerization ability [31,32,33]. This effect was examined using nanoparticle tracker (Figure 1G). Consistent with other studies, WNPRBD protein alone can form aggregates/oligomers in solution with an overall positive surface change. On the other hand, smaller particles were formed with WNPRBD-R266, with an overall negative charge. These WNPRBD-R266 particles were certainly due to the association of protein and RNA since RNA alone did not form detectable particles, and the surface charge of WNPRBD-R266 was opposite to the protein alone. Interestingly, the sizes of nanoparticles formed were mostly uniform and not in a wide range of different sizes. This implied that viral nucleoprotein may have specific mechanisms to form ordered structures for virus genome packaging, as discussed in other studies [34]. The RNA-binding properties of WNPRBD were also determined by gel shift assay (Figure 1I) and fluorescence polarization assay (Figure 1J) and found to be similar to full-length nucleoproteins.

The immunogenicity of WNPRBD-R266 RNP vaccine was then proven in subsequent mice experiments. WNPRBD-R266 can effectively induce mucosal IgA and IgG and systemic serum IgG (Figure 2A–D). As shown in Figure 2, the RNA length did affect the vaccine’s immunogenicity. WNPRBD-R20 elicited serum antibodies but failed to induce mucosal IgA. Although various lengths of RNAs have been studied for different therapeutic purposes, our data shows that shorter RNA is ineffective in triggering mucosal immunity during intranasal vaccination. However, shorter RNA does not mean a weaker immunostimulant. As seen in our cytokine analysis (Figure 2E–G), R20 and R266 complexes were both able to induce strong interferon response and chemokine signals. Both model RNAs are immunostimulative, yet the immunization outcome presents a large discrepancy in IgA induction. This hinted that the length of the RNA in the RNP vaccine must be carefully designed. Although common knowledge infers that viral nucleoproteins bind to nucleic acids through charge interaction, they may still have a higher affinity to viral sequences. Also, the mechanism behind the robust IgA induction in WNPRBD-R266 vaccination but not -R20 definitely requires further investigation. In summary, WNPRBD-R266 intranasal immunization in mice effectively triggers systemic and mucosal antibody responses.

The antibody response from WNPRBD-R266-vaccinated mice was further elucidated by looking at different RNA dosage, antibody subclasses, and neutralization ability (Figure 3). It was found that 25 µg of R266 was the optimal dose for strong mucosal activation. The vaccine also induces a balanced IgG1 and IgG2b response and functional neutralizing antibodies. The antigen-specific T cell responses were also examined (Figure 4), where robust CD4+ and CD8+ T cell activation was identified. WNPRBD-R266-vaccinated mice were also protected from the live SARS-CoV-2 virus challenge (Figure 5), showing sterilizing immunity. We also observed CD4+ T cell response in the lung, but marginal CD8+ (Appendix A). These data show that the WNPRBD-R266 RNP vaccine induces comprehensive immunity in the preclinical animal model.

An interesting finding presented in this study is the importance of complex formation for WNPRBD-R266 (Figure 3N–T). A mixture of RBD-only R266 was compared to the RNP complex WNPRBD-R266. It was impossible to generate a nucleoprotein mutant that does not bind RNA. The “mixture” vaccination was incapable of inducing mucosal IgA in mice and failed to elicit a balanced immune response. In contrast, the WNPRBD-R266 complex induced a more balanced immune response similar to the intramuscular BioNTech mRNA vaccine in BALB/c mice. These data provided important insights that RNP complex immunization differs entirely from simply mixing proteins and adjuvants. One possible explanation for the difference may be that if the R266 is acting as an adjuvant mixed in, immune cells such as macrophages and monocytes are first triggered and activated, releasing chemokines to further attract other antigen-presenting cells (APCs) to the site and engulf the protein antigen for presentation. This, however, does not resemble a true infection where viruses enter cells as a whole. But for the WNPRBD-R266 RNP complex, both the immunostimulant and protein antigen are co-delivered at the same instance as uptake by macrophages and APCs, which is similar to a true virus infection. We hypothesize that this subtle difference between “activate-then-present” and “co-delivery” is the key to the effectiveness of WNPRBD-R266 RNP. The detailed mechanism should be further elucidated in future studies.

We also utilized scRNA-seq techniques to examine airway-resident immune cells in mice (Figure 6). Interestingly, the profiles of Naïve and Prime mice were similar. This indicates that the first dose contributes to the priming effect for primary immunization. A second dose should be administered as boost to activate immune memory, resulting in strong immunity. At 7 days post-2nd dose, B cells, T cells, NK cells, and NKT clusters were enriched, indicating proliferation and activation of immune memory primed by Prime. The Boost dose effectively activated all major immune cell populations in the airway compartment. Moreover, innate-like T cells were identified in both MMN and T cell clusters. These cells are recognized as having critical roles in antitumor immunity [35]. Innate CD8+ T cells also participate in controlling persistent, long-term viral infections [36]. In humans, innate-like T cells are tissue-resident lymphocytes that control proinflammatory cytokines and contribute to rapid antimicrobial responses [37]. All in all, the scRNA-seq data provided high-resolution evidence for the robust immune activation by WNPRBD-R266 and highlighted the transition state from prime to boost. This dataset provided important insights into the future mechanistic study of the vaccine on immune cell activation.

It is important to state that in this study all animal immunizations were performed using wild-type BALB/c mice, which is not the best animal model for SARS-CoV-2 infection studies. We made reference to recent studies and chose Beta B.1.351 for infection experiments in order to generate reasonable data and evaluate WNPRBD-R266 efficacy [38]. In future follow-up studies, other strains such as hACE2 transgenic mice should be used to fully demonstrate vaccine protection against SARS-CoV-2 infection.

The potential of WNPRBD-R266 can be extended to other viruses as well. Biologically, all viruses contain surface protein epitopes that act as neutralizing targets and nucleoproteins that package their genome. Nucleic acids such as RNA are essential to all live viruses. This indicates that this design can be applied to other kinds of viruses. As a conceptual experiment, we designed and produced an influenza A version of the protein-R266 vaccine, simply by replacing the SARS-CoV-2 spike RBD with the H1N1/pdm09Md HA head (Appendix A). Mice were immunized with two doses and antibody responses were evaluated. Interestingly, the same magnitude of systemic response and induction of mucosal antibodies were observed. This shows the modularity of protein-R266 and that it can be established as a novel vaccine platform to produce vaccines for different kinds of viruses. The RNP vaccine in this study consists of non-infectious proteins and ssRNA as an adjuvant that induces minimal inflammatory cytokines. This is promising for a safe and robust nasal vaccine. Regarding the feasibility of production for clinical use, both protein and RNA manufacturing are highly mature in the industry, and therefore can be integrated into current platforms for large-scale production. The components can also be lyophilized to enhance their stability and hence potentially reduce the need for ultra-cold temperature transportation. The stability and pharmacokinetics of the WNPRBD-R266 complex should be investigated thoroughly in future studies and to facilitate the transition to clinical application.

## 5. Conclusions

In summary, WNPRBD-R266 is a protein nasal vaccine comprising a protein-RNA complex. Systemic and mucosal immunity was significantly induced in the preclinical mouse model. Vaccinated mice also generated robust antigen-specific T cell responses and were protected against a live virus challenge. scRNA-Seq analysis revealed the importance of prime–boost vaccination for primary immunization and provided a top-down perspective on bronchioles and the lung-resident immune cell profile. WNPRBD-R266 is, therefore, a promising nasal vaccine candidate and further clinical evaluation is very much required.

## Figures and Tables

**Figure 1 vaccines-11-01550-f001:**
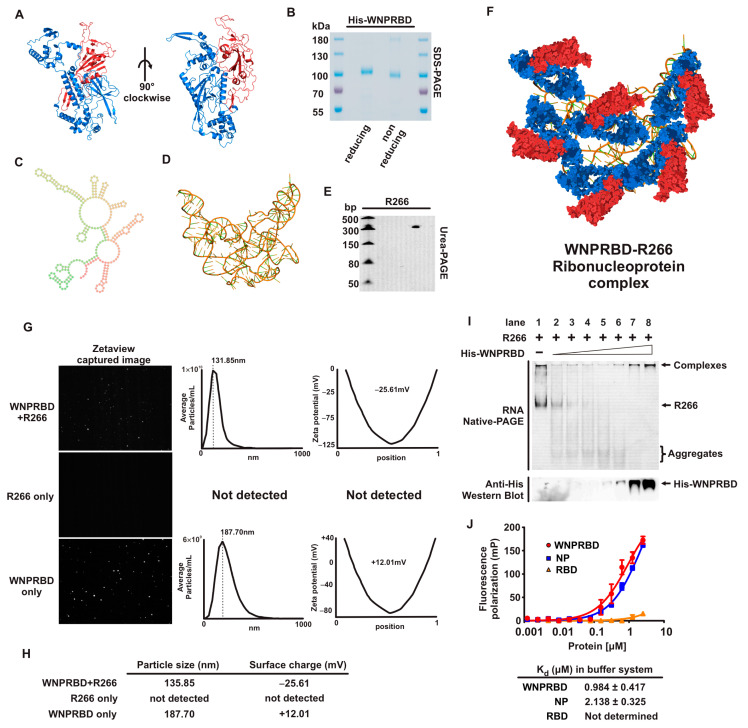
The novel ribonucleoprotein complex. (**A**) 3D-structure of the protein component WNPRBD, predicted using Robetta web server. Red domain: RBD; Blue domain: whole nucleoprotein (WNP). (**B**) SDS-PAGE analysis of WNPRBD protein. (**C**) Secondary structure prediction of the RNA component R266, predicted by RNAfold server. (**D**) 3D-structure prediction of R266, predicted by RNAcomposer server. (**E**) Urea-PAGE analysis of R266 RNA. (**F**) Conceptual illustration of WNPRBD-R266 ribonucleoprotein complex. (**G**) Nanoparticle tracking and analysis using Zetaviewer. (**H**) Summary of nanoparticle sizes and Zeta potential. Data shown represented average of 3 independent measurements. (**I**) RNA gel shift assay with WNPRBD and R266. (**J**) Fluorescence polarization assay with WNPRBD, full-length nucleoprotein, and RBD protein.

**Figure 2 vaccines-11-01550-f002:**
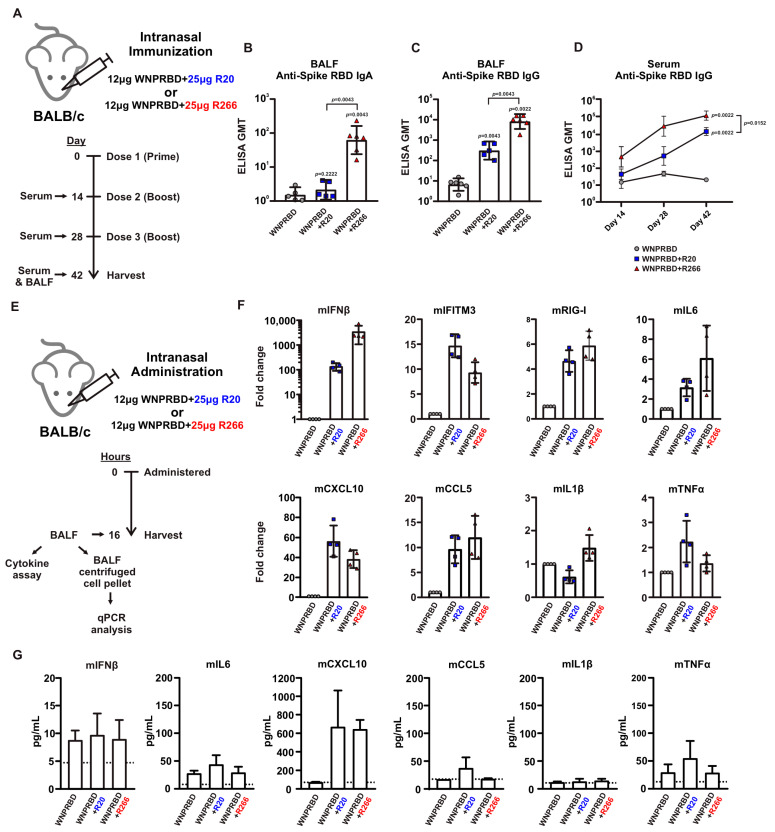
WNPRBD-R266 complex is a self-adjuvanted ribonucleoprotein nasal vaccine. (**A**) Intranasal immunization of ribonucleoprotein in mice. An amount of 12 μg WNPRBD was mixed with either 25 μg R20 or R266 to form ribonucleoprotein vaccine. Mice (*n* = 6) were immunized intranasally with 3 doses of vaccines in 14 days interval between each dose. Sera and bronchioalveolar fluid (BALF) were harvested at experiment endpoint and further assayed by ELISA. (**B**–**D**) Mice BALF anti-RBD IgA at day 42, BALF anti-RBD IgG at day 42, and serum anti-RBD at day 14, 21, 42 determined by ELISA. Data was represented as geometric mean titer (GMT), 95% confidence interval. Statistical significances were determined using Mann–Whitney test, where *p* < 0.05 was considered significant, comparing WNPRBD-R266 or -R20 to protein-only, or comparing R20 and R266, respectively. Exact *p* values were shown. (**E**) To determine the immunostimulative effect of the RNA component, WNPRBD-R20 or -R266 were intranasally administered to mice (*n* = 4). At 16 h post-administration, BALFs were harvested, centrifuged, and the resulting cell pellets were subjected to RNA extraction. BALF cell pellet RNAs were analysed by qPCR for cytokine transcript expressions, while BALFs were analyzed by cytokine assay to determine cytokine protein expression. (**F**) qPCR analysis of cytokines and chemokines transcript expression in BALFs flushed cells. (**G**) Bead-based cytokine assay of cytokines and chemokines protein expression in BALFs collected from the same mice (*n* = 4). Data of qPCR and cytokine assay were represented as sample mean, with sample standard deviation (SD).

**Figure 3 vaccines-11-01550-f003:**
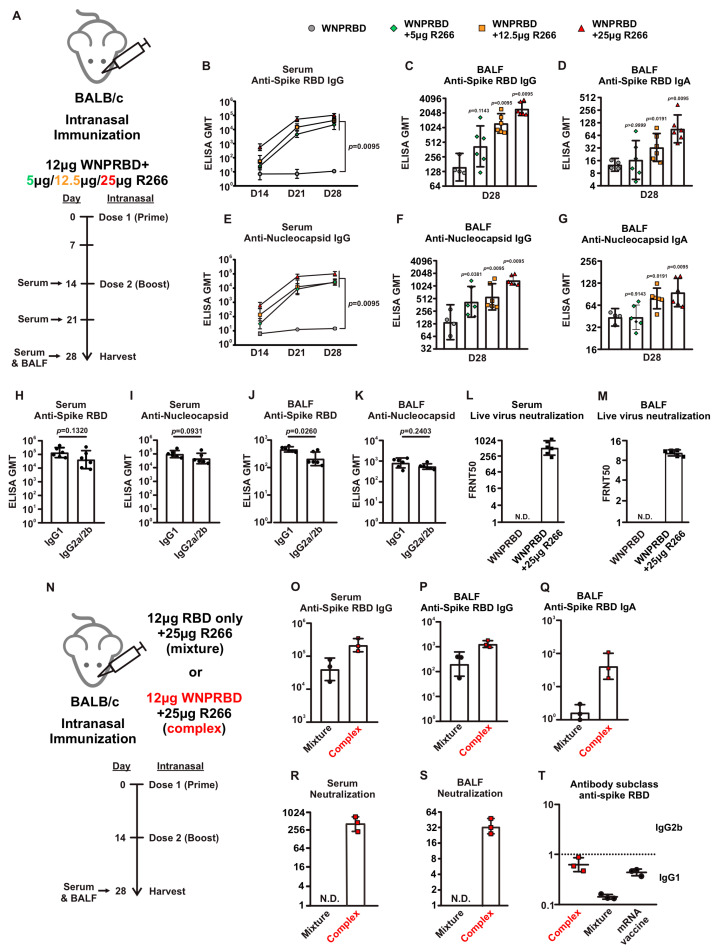
Intranasal immunization of WNPRBD-R266 ribonucleoprotein induces strong and balanced immune response. (**A**) To determine optimal dosage, mice (*n* = 6) were intranasally immunized with WNPRBD-R266 ribonucleoprotein with different dose of R266 (5/12.5/25 µg). Mice were given 2 doses of vaccine, with 14 days interval, and sera and BALF were harvested at endpoint. (**B**–**D**) Anti-spike RBD antibody responses in mice sera and BALFs at different R266 doses. (**E**–**G**) Anti-nucleoprotein antibody responses in mice sera and BALFs at different R266 doses. (**H**–**K**) Anti-spike RBD and Anti-nucleoprotein IgG1 and IgG2a/2b levels in mice sera (**H**,**I**) and BALFs (**J**,**K**) with 12 µg WNPRBD-25 µg R266. Antibodies titers were determined by ELISA, data was represented as GMT, 95% confidence interval. Statistical significances were determined using Mann–Whitney test, where *p* < 0.05 was considered significant, comparing WNPRBD with different R266 doses to protein-only, or comparing IgG1 and IgG2a/b, respectively. Exact *p* values were shown. (**L**,**M**) Neutralizing antibody levels in sera (**L**) and BALFs (**M**) determined by FRNT with 12 µg WNPRBD-25 µg R266. N.D.: Not Detected. (**N**) To investigate the importance of complex formation, 12 µg recombinant spike RBD proteins were mixed with 25 µg R266 and intranasally administered to mice (*n* = 3). (**O**–**Q**) Comparison of antibody levels induced by 12 µg RBD-only 25 µg R266 mixture or 12 µg WNPRBD-25 µg R266 complex in sera (**O**) and BALF (**P**: IgG, **Q**: IgA). (**R**,**S**) Neutralization assay comparing mixture and complex (sera: R, BALF: S). (**T**) IgG2b/IgG1 antibody ratio in sera immunized with either WNPRBD-R266 complex, RBD-only-R266 mixture and mRNA vaccine.

**Figure 4 vaccines-11-01550-f004:**
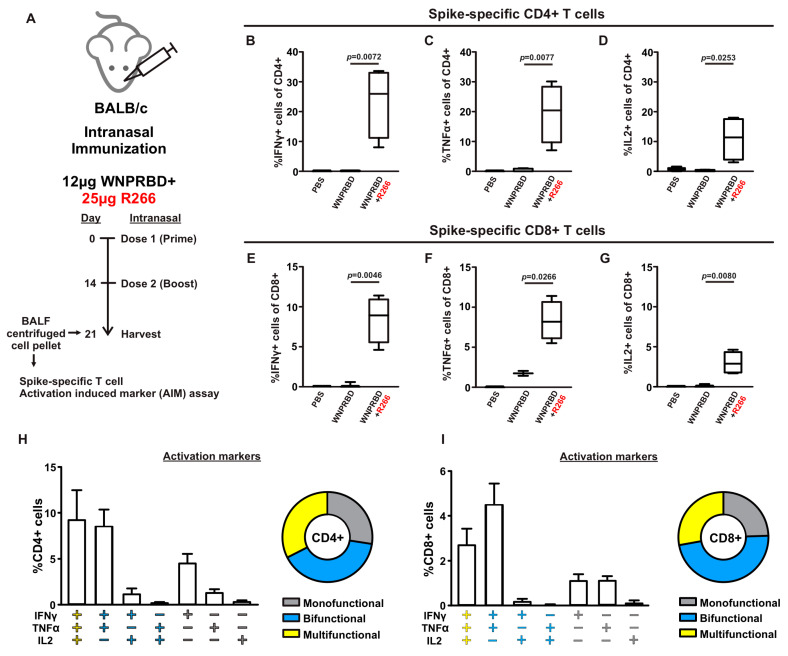
WNPRBD-R266 induces robust spike-specific resident T-cells in mice BALF. (**A**) Mice (*n* = 4) were intranasally immunized with WNPRBD-R266, at 2 doses, and BALF flushed cells were harvested at day 7 post-2nd dose. Cells were then stimulated with SARS2 spike peptides pool determine activation of antigen-specific T cells. (**B**–**D**) Spike-specific CD4+ T cells expressing activation markers IFNγ, TNFα, and IL2. (**E**–**G**) Spike-specific CD8+ T cells expression activation markers. Summary of percentage of CD4+ (**H**) and CD8+ (**I**) T cells expressing monofunctional, bifunctional, and multifunctional activation markers. Statistical significances were determined using an unpaired *t*-test, where *p* < 0.05 was considered significant, comparing protein-only and WNPRBD-R266.

**Figure 5 vaccines-11-01550-f005:**
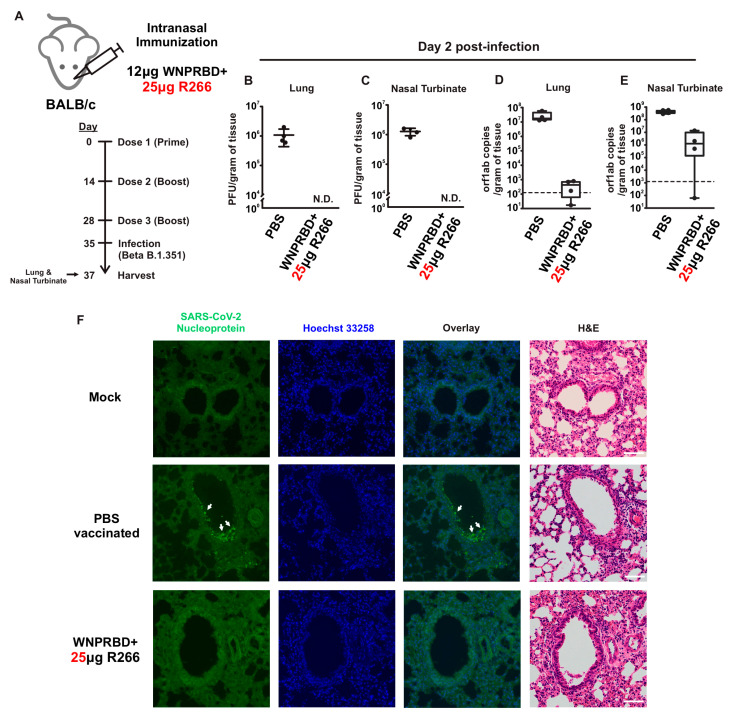
WNPRBD-R266 ribonucleoprotein vaccination protects mice from SARS-CoV-2 infection and induces sterilizing immunity. (**A**) Mice (*n* = 4) were intranasally immunized with 3 doses of WNPRBD-R266 and challenged with Beta B.1.351 virus at day 7 post-3rd dose. Lungs and nasal turbinates were harvested at day 2 post-infection. (**B**,**C**) Viral titer analyzed by plaque assay. Data was shown as sample mean with sample SD. N.D.: Not Detected. (**D**,**E**) Viral loads analyzed by qPCR. Data were shown as box chart with minimum and maximum value. (**F**) Immunohistology staining for nucleoprotein positive cells. Scale: 100 µm.

**Figure 6 vaccines-11-01550-f006:**
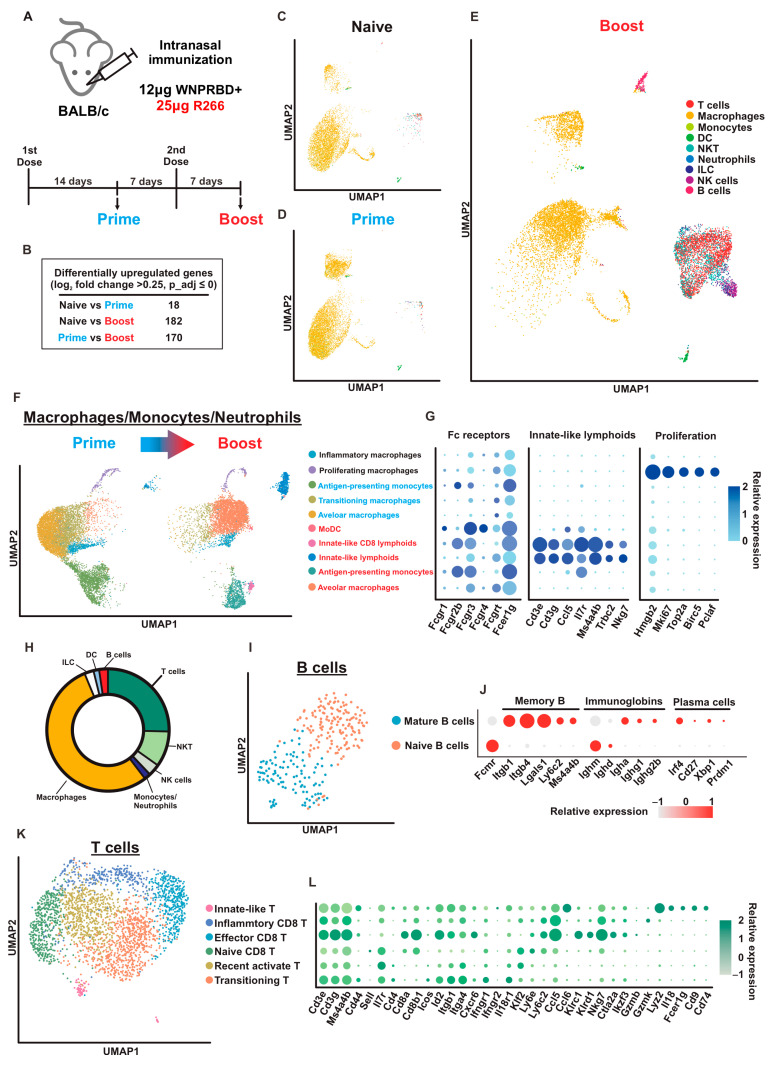
High resolution immune cell profiling in WNPRBD-R266 intranasal vaccinated mice BALF by single-cell RNA sequencing. (**A**) Mice (*n* = 1 for each sample) were immunized with 2 doses of WNPRBD-R266. BALF-flushed cells were harvested at 7 days before 2nd dose (Prime) and 7 days after 2nd dose (Boost). Cells were then subjected to single-cell RNA sequencing. (**B**) Differentially upregulated gene expression of Naïve, Prime and Boost samples. (**C**–**E**) UMAP reduction of Naïve, Prime and Boost clusters. Respective cell identities were highlighted in different colors. (**F**) UMAP plot of macrophage, monocyte, and neutrophil (MMNs) reclustering. (**G**) Dotplot representing gene markers expression within MMNs cluster. Prime-specific clusters are highlighted in blue, Boost-specific clusters are red, and clusters appearing in both Prime and Boost are in black. (**H**) Donut plot of cell type populations in the Boost sample. (**I**) UMAP plot of Boost B cell reclustering. (**J**) Differential gene markers within the Boost B cell cluster. (**K**) UMAP plot of Boost T cell reclustering. (**L**) Dotplot representing differential gene marker expression in the Boost T cell cluster.

## Data Availability

Data supporting this study’s findings are available in this paper. Appendix A are available upon request. The raw data of scRNA sequencing were deposited into NCBI Sequence Read Archive (BioProject number PRJNA995586).

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
