# Peer review of "An RNA-Scaffold Protein Subunit Vaccine for Nasal Immunization"

_vaccines, 2023, doi:10.3390/vaccines11101550_

Round 1

Reviewer 1 Report

The authors proposed the concept of an intranasal vaccine concept of RNP mixed with surface spike protein, adapter nucleoprotein and RNA as adjuvant as a vaccine for respiratory virus infection.
Using SARS-CoV-2 as a model, the RNP was made by mixing the RBD of the S protein with a full-length N-fused protein and 266 nt of RNA in the 5' UTR. By administering this RNP to mice, IgA and IgG in BALF and blood were induced. Both BALF and blood contained neutralizing antibodies. Analysis of various cytokines showed that inflammation was minimized. Furthermore, infection of immunized mice with SARS-CoV-2 resulted in no detectable infectious particles or very low levels of gRNA in the lungs of the immunized mice. The authors further demonstrated its superiority based on immunological and multifaceted analyses, including scRNA-seq analysis. From the above, this paper is well written and the research results are novel. However, the following points must be addressed.

Minor comments

1.     The authors did not examine the extent to which the RNP affects antigen testing from the nasal cavity after immunization, nor did they describe their predicted false positives. It would be more valuable if the N antigen could be demonstrated by staining on pathological tissue thin sections of the nasal turbinate over time post-immunization.

2.     I had difficulty understanding the ratio of RNA to the fusion protein included as RNP. It states that 12ug of WNPRBD was used. For example, if it says 25ug of RNP was used, does that mean 12ug of WNPRBD and 13ug of RNA? Please clarify.

3.     Please state the volume of the viral solution used in the SARS-CoV-2 challenge experiment: how many microliters of solution is 1x10^5PFU dissolved in?

4.     The mice used in this study were normal BALB/c, and the virus used in the challenge experiment does not seem to be a mouse-adapted virus. Although the difference from the non-immune group is sufficient, this RNP should be evaluated in future animal model systems in which pneumonia and lethality can be assessed. This point should be mentioned somewhere.

5.     Fig1A. Which is RBD and which is N, shown in red and blue?

6.     The Fig 2G bar has no symbols, unlike the other Fig bars. Please state how many mice were used in the experiment.

7.     There is a typo in Fig 5 legend." SARS2"

8.     What does the ND in Figs 3 and 5 mean?

9.     In some places it is written in units of bp even though it is about ssRNA. Should it be nt or base?

10.   I think the A-K in Fig 6 and its legend does not seem to fit. Please check and correct.

Author Response

Reply to reviewer:

Comment #1:

Using SARS-CoV-2 as a model, the RNP was made by mixing the RBD of the S protein with a full-length N-fused protein and 266 nt of RNA in the 5' UTR. By administering this RNP to mice, IgA and IgG in BALF and blood were induced. Both BALF and blood contained neutralizing antibodies. Analysis of various cytokines showed that inflammation was minimized. Furthermore, infection of immunized mice with SARS-CoV-2 resulted in no detectable infectious particles or very low levels of gRNA in the lungs of the immunized mice. The authors further demonstrated its superiority based on immunological and multifaceted analyses, including scRNA-seq analysis. From the above, this paper is well written and the research results are novel. However, the following points must be addressed.

Address #1:

Thank you very much for your comment. We hope that our RNP vaccine can overcome the low immunogenicity of protein nasal vaccines and eventually can benefit the general public to prevent respiratory diseases.

Comment #2:

The authors did not examine the extent to which the RNP affects antigen testing from the nasal cavity after immunization, nor did they describe their predicted false positives. It would be more valuable if the N antigen could be demonstrated by staining on pathological tissue thin sections of the nasal turbinate over time post-immunization.

Address #2:

Thank you for the comment and we fully agree on your advice. In this study, we focus on developing the RNP vaccine platform and test its efficacy in animal models. In the follow up studies that we are currently working on, we begin to investigate the pharmacokinetics of RNP vaccines. Similar to your advice, we are currently trying to stain and determine the presence of RNP in mice head, nasal cavity, nasal turbinate, and lung post-immunization. We hope to provide more insights in future publications.

Comment #3:

I had difficulty understanding the ratio of RNA to the fusion protein included as RNP. It states that 12ug of WNPRBD was used. For example, if it says 25ug of RNP was used, does that mean 12ug of WNPRBD and 13ug of RNA? Please clarify.

Address #3:

Thank you for pointing out and we apologize for the confusion.

In Figure 2, we used 12mg protein + 25mg R266 for immunization.

In Figure 3A-G, we tested 12mg protein + different dosage of R266.

In Figure 3H-M, since we found that 12mg protein + 25mg R266 gave the best response, this was the condition used for these data.

In Figure 3N-T, 4, 5, 6, 12mg protein + 25mg R266 was used for all these data.

In summary, all immunizations were performed with 12mg protein, then we tried to clarified the amount of R266 used in each Figure. Sorry for the confusion and we have modified the figures so that all will show how much protein or R266 were used.

Comment #4:

Please state the volume of the viral solution used in the SARS-CoV-2 challenge experiment: how many microliters of solution is 1x10^5PFU dissolved in?

Address #4:

Thank you for pointing out. The volume used for intranasal inoculation of virus was 20mL. We have updated the methods section to clarify.

Comment #5:

The mice used in this study were normal BALB/c, and the virus used in the challenge experiment does not seem to be a mouse-adapted virus. Although the difference from the non-immune group is sufficient, this RNP should be evaluated in future animal model systems in which pneumonia and lethality can be assessed. This point should be mentioned somewhere.

Address #5:

We fully agree on your interpretation. Wild-type BALB/c mice do not respond well in SARS-CoV-2 virus challenge experiments. We are trying to acquire hACE-2 BALB/c transgenic mice for future studies. We have modified the manuscript and added information on this in the “Discussion” section.

Comment #6:

-Fig1A. Which is RBD and which is N, shown in red and blue?

-The Fig 2G bar has no symbols, unlike the other Fig bars. Please state how many mice were used in the experiment.

-There is a typo in Fig 5 legend." SARS2"

-What does the ND in Figs 3 and 5 mean?

-In some places it is written in units of bp even though it is about ssRNA. Should it be nt or base?

-I think the A-K in Fig 6 and its legend does not seem to fit. Please check and correct.

Address #6:

Thank you for pointing out. We apologize for the mistakes and confusion. We have modified the manuscript and addressed these issues. For Figure 2G, the BALFs were collected from the same mice in Figure 2F, thus n=4. ND refers to Not Detected. For ssRNA, “nt” should be used instead of “bp”.

Reviewer 2 Report

This manuscript developed a self-adjuvant protein-RNA ribonucleic acid protein vaccine and found it to be an effective nasal vaccine in mice and SARS-18 CoV-2 infection models. The main structure of the vaccine is composed of spines, protruding RBD(as antigen), nuclear protein (as connector) and ssRNA(as adjuvant and rna scaffold). According to the results presented in the manuscript, the vaccine can effectively induce mice to secrete antigen IgA, neutralize antibodies and activate multifunctional T cells. It also provides immunity against the virus. In addition, the manuscript also used the RNA-SEQ technique to analyze the corresponding immune cells, and obtained a good primary immune effect. From a model point of view, the RNA ribonucleoprotein vaccine proposed in this manuscript looks like a very promising approach. The design of this work sounds very interesting, but there are some questions to be explained in this paper that preclude its publications in this form.

1If this vaccine can go out of the clinic, how do you consider its production, transportation and preservation?

2Page 3, section 2.6,, the mice were already 8-10 weeks old when they were selected, and then after 14 weeks of contact modeling, the weekly age was 22-24 weeks, the mice were older, so are the results of the test applicable to all ages?

3Page 3, section 2.6, the cycle of viral infection in mice is clear, but the time and interval between vaccinations is not clearly written.

4Page 11, line 115, please re-write the first sentence and check the syntax of the whole text.

5The education of Fig1B was damaged in the supplementary literature. Can you provide the complete glue map? This will make your essay more complete.

6Page12Fig 5F, Please complete the scale in the drawing.

There are some minor editing of English language required.

Author Response

Reply to reviewer:

Comment #1:

This manuscript developed a self-adjuvant protein-RNA ribonucleic acid protein vaccine and found it to be an effective nasal vaccine in mice and SARS-18 CoV-2 infection models. The main structure of the vaccine is composed of spines, protruding RBD(as antigen), nuclear protein (as connector) and ssRNA(as adjuvant and rna scaffold). According to the results presented in the manuscript, the vaccine can effectively induce mice to secrete antigen IgA, neutralize antibodies and activate multifunctional T cells. It also provides immunity against the virus. In addition, the manuscript also used the RNA-SEQ technique to analyze the corresponding immune cells, and obtained a good primary immune effect. From a model point of view, the RNA ribonucleoprotein vaccine proposed in this manuscript looks like a very promising approach. The design of this work sounds very interesting, but there are some questions to be explained in this paper that preclude its publications in this form.

Address #1:

Thank you very much for your comment. We hope that our RNP vaccine can overcome the low immunogenicity of protein nasal vaccines and eventually can benefit the general public to prevent respiratory diseases. We agree that there are many aspects that require attention if it is to be brought into clinical use, and we are working hard to address them.

Comment #2:

If this vaccine can go out of the clinic, how do you consider its production, transportation and preservation?

Address #2:

Thank you for raising these comments. In terms of production and manufacturing, we foresee not too many obstacles since protein and RNA manufacturing are very mature within the industry. For transportation and storage, our preliminary results showed that lyophilized form of protein and short ssRNA are stable in room temperature. The two components, protein and RNA, can be mixed on site. We are currently performing more testing to confirm these observations. We have added some information on this in the manuscript to enrich the discussion.

Comment #3:

Page 3, section 2.6,, the mice were already 8-10 weeks old when they were selected, and then after 14 weeks of contact modeling, the weekly age was 22-24 weeks, the mice were older, so are the results of the test applicable to all ages?

Address #3:

Thank you for the comment. For mouse immunization and BALF harvesting, mice were immunized with 2 doses of vaccine, 14 days apart. Thus mice were sacrificed at age of 12-14 weeks old. For animal infection, mice were immunized with the same plan, and perform infection at age of 14 weeks old. We have modified the method section to make it more clear.

Comment #4:

Page 3, section 2.6, the cycle of viral infection in mice is clear, but the time and interval between vaccinations is not clearly written.

Address #4:

Thank you again. Following Comment & Address #3, we have modified the method section to clarify.

Comment #5:

Page 11, line 115, please re-write the first sentence and check the syntax of the whole text.

Address #5:

Sorry for the mistake. It seems to be a formatting problem and we have fixed it in the revised manuscript.

Comment #6:

The education of Fig1B was damaged in the supplementary literature. Can you provide the complete glue map? This will make your essay more complete.

Address #6:

Thank you for pointing out. We have revised the figure for a better gel image for Fig1B.

Comment #7:

Page12,Fig 5F, Please complete the scale in the drawing.

Address #7:

Thank you for pointing out. We have updated the scale in the figure.

Comment #8:

There are some minor editing of English language required.

Address #8:

Thank you for the advice. We have revised the English language in the manuscript.

Reviewer 3 Report

In the current study, Lam et al. has designed an RNP based intranasal COVID vaccine and demonstrated its effectiveness in stimulating strong and protective immune responses against SARS-CoV2. The authors have performed extensive assays including single cell sequencing to describe the robustness of the vaccine in stimulating antibody, cellular and cytokine responses which is quite impressive. The vaccine design is innovative and has potential, hence, RBDs of latest SARS-CoV2 variants with relevant mutations would be worth incorporating to test in future for complex stability and efficacy. Just a minor comment about the Discussion in the paper, several intranasal COVID vaccines (phage-, RNA-, vector-based) have been tested in preclinical/clinical studies recently which should be referenced and discussed in Discussion.Advantages of this vaccine design over existing should also be clearly described. Overall, it is an interesting study.

Author Response

Reply to reviewer:

Comment #1:

In the current study, Lam et al. has designed an RNP based intranasal COVID vaccine and demonstrated its effectiveness in stimulating strong and protective immune responses against SARS-CoV2. The authors have performed extensive assays including single cell sequencing to describe the robustness of the vaccine in stimulating antibody, cellular and cytokine responses which is quite impressive.

Address #1:

Thank you very much for your comment. We hope that our RNP vaccine can overcome the low immunogenicity of protein nasal vaccines and eventually can benefit the general public to prevent respiratory diseases.

Comment #2:

The vaccine design is innovative and has potential, hence, RBDs of latest SARS-CoV2 variants with relevant mutations would be worth incorporating to test in future for complex stability and efficacy.

Address #2:

Thank you for your suggestion and we fully agree. The different RBD variant for RNP vaccine can enhance the protection against more updated virus variants. Therefore, we proposed the RNP vaccine to be a vaccine platform and not limit to one virus. We have also included the influenza A H1N1 RNP as an example in the supplementary figure 2.

Comment #3:

Just a minor comment about the Discussion in the paper, several intranasal COVID vaccines (phage-, RNA-, vector-based) have been tested in preclinical/clinical studies recently which should be referenced and discussed in Discussion. Advantages of this vaccine design over existing should also be clearly described. Overall, it is an interesting study.

Address #3:

Thank you for the suggestion. Our vaccine is advantageous because there is no protein nasal vaccine available at the moment. Our RNP vaccine consists of non-infectious proteins and RNA as adjuvant that induce minimal inflammatory cytokines. We have added the discussion of other intranasal vaccines and clinical trials in the revised manuscript accordingly.